# Developing a threat risk register based on the IUCN threat hierarchy for five tropical Important Plant Areas in Guinea

Charlotte Couch[1,2,3], Faya Julien Simbiano[1], Mamadou Diawara[4], Edgar François Loua[5], Léonce Mamy[6], Martin Cheek[3,7] and Sékou Magassouba[1,3]

[1] Herbier National de Guinée, Conakry, Guinea
[2] Royal Botanic Gardens Kew, Richmond, Surrey, United Kingdom
[3] IUCN Species Survival Commission, West Africa Plant Red List Authority, Gland, Switzerland
[4] Guinée Ecologie, Conakry, Guinea
[5] Conakry, Guinea
[6] Centre Forestier N'Zérékoré, N'Zérékoré, Guinea
[7] Herbarium, Royal Botanic Gardens, Kew, Richmond, Surrey, United Kingdom

Corresponding author
Charlotte Couch, c.couch@kew.org

## ABSTRACT

A pilot study to develop a threat risk register for Tropical Important Plant Areas in Guinea using the IUCN threat hierarchy is outlined. Guinea lost 92% of its total original forest before the end of the 20th Century. In addition, in the Guinée Forestière region alone, a further 25% of the remaining forest has been lost between 2000 and 2018, primarily driven by agriculture. One of the obstacles to effective protected area management in Guinea is the lack of quantitative measurements of the characteristics and location of the threats. Data was collected from five areas in Guinée Forestière to create individual risk registers for mapping and monitoring threats. The results show that the biggest threat is from agriculture, followed by biological resource use and intrusions and human disturbance. The level of threat of agriculture varies between sites but is the greatest threat at Mt Béro and Southern Simandou Mountains, though results could be skewed by sampling density. Further training of conservators and ecoguards on identification and classification of threats is needed to ensure consistency of recording across areas. This is a novel technique for recording and quantifying threats to plants in protected areas in Africa as no equivalent has been found during the course of this research. This tool has potential uses, both nationally and internationally, to improve monitoring of threats to rare plants and the forest landscape and can feed into IUCN Red List species and ecosystem assessments, as well as Protected Area Management Effectiveness systems.

# INTRODUCTION

In this work, we explore the creation of a risk register for Tropical Important Plant Areas (TIPAs) identified in Guinea in 2019 by *Couch et al. (2019)*. The risk register is as a means

of providing an efficient way of gathering data for monitoring, mitigation and forward policy planning.

In Guinea, there are general known risks to the flora especially forests, as already outlined in management and development plans (*MEDD, 2021*), but on-the-ground implementation of mapping and monitoring of these risks is still lagging. Conservators and ecoguards patrol the forests for signs of poaching and illegal tree cutting or clearing using the SMART (Spatial Monitoring and Reporting Tool) system (https://smartconservationtools.org/), but other smaller scale threats go unrecorded. Moreover, interpreting what is a threat can also be difficult for people on the ground when no definitions are provided or if there is a lack of knowledge around the ecology of plant species threatened. Using a unified classification of threats with specific definitions, such as the lexicon developed by *Salafsky et al. (2008)* and now under the management of the IUCN Classification Schemes Working Group, enables threats to be categorised and analysed across sites and between countries (*BGCI, 2021*).

Risk or threat registers are used to identify threats in project or organisational management (*Royal Botanical Gardens, Kew, 2021*; https://www.stakeholdermap.com/risk/register-commonproject-risks.html (accessed on 07 July 2022); *UNDP, 2023*); however, a risk register framework can provide a useful way to identify, record and manage threats in a wide range of scenarios. *Mace et al. (2015)* looked at using this concept to create a risk register for Natural Capital. Although not perfect, the process of gathering the information needed for the register helped to indicate areas which could, with more data collection and research, produce a robust and relevant policy level tool. A pilot study was undertaken to create preliminary risk registers for five TIPAs which correspond to the Key Biodiversity Areas (KBAs) (*IUCN, 2016a*) of Guinée-Forestière as defined by the Critical Ecosystem Partnership Fund Biodiversity Hotspots of the Guinean Forests of West Africa (*CEPF, 2015*).

The study area includes the KBA/Tropical Important Plant Areas (TIPAs) of Mont Béro, Diécké Classified Forests, the Southern Simandou mountains which includes the Pic de Fon Classified Forest, the Guinean part of the UNESCO World Heritage Area of Mts Nimba and the "Man and Biosphere Reserve" of the Massif de Ziama, in the south-east of Guinea (*i.e.,* Guinée Forestière, Fig. 1). These sites contain the largest remnants of lowland and submontane forest in Guinea and are highly important for the conservation of many threatened and endemic plant species (*Couch et al., 2019*); however, this is not always reflected in the management and development plans (PAG) (*MEDD, 2021*; *MEDD, 2022a*; *MEDD, 2022b*). It is also important to note that Classified Forests (or Forêts Classées in French) were originally established for sustainable timber production, though protection of wildlife was also taken into some consideration. Those created before independence were largely abandoned and many were cleared by farmers for agriculture (*Brugière & Kormos, 2009*; *IUCN/PACO, 2008*). Some of the remaining classified forests have become incorporated into the protected area network in Guinea, for example most of those in this study, but this is not the case for the majority nationally. Until recently, written management plans for Mt Béro and Diécké consider the general site-based threats but did not consider the diversity of, or specific threats to, the flora. This prompted development of (the first nationally) two Conservation Action Plans for plants of Mt Béro and Diécké

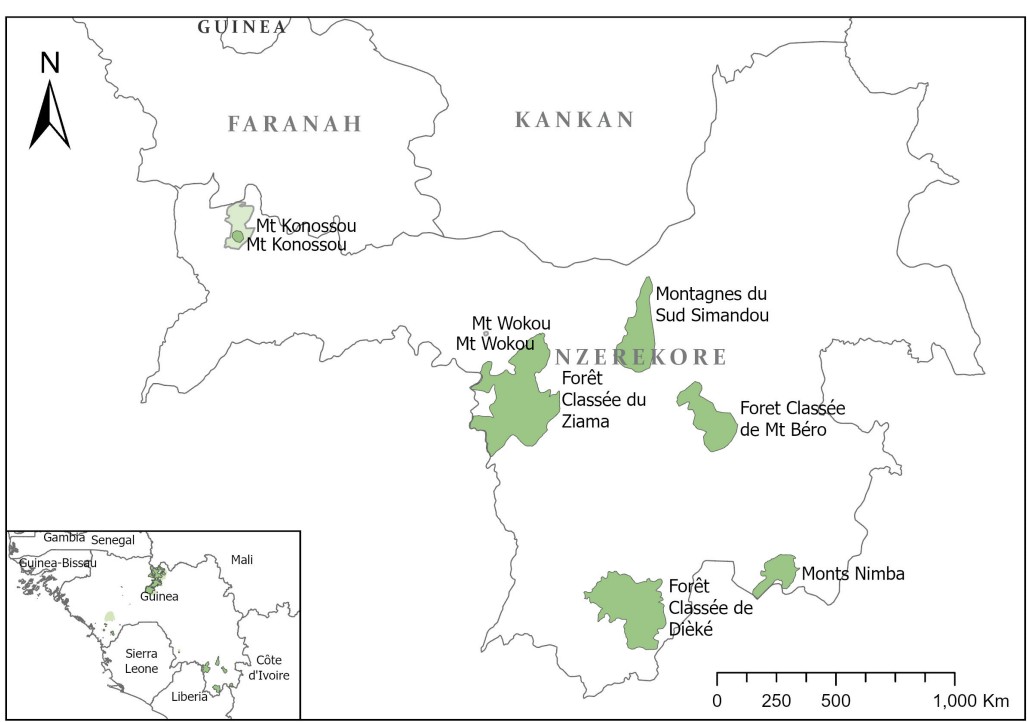

**Figure 1 Map of tropical important plant areas in Guinée-Forestière, N'Zérékoré governorate.**

(*Diaby, Couch & Simbiano, 2022*; *Couch & Simbiano, 2021*). This led to the realisation that a method to record and quantify these threats is needed.

Guinea has an estimated rural population of 63%, increasing annually by 2.1% (*World Bank, 2021*). Many of these rural villages depend on the natural habitats such as forests for medicines, food and materials, often leading to detrimental effects upon those ecosystems (*MEDD, 2022b*). A local study of indigenous socioeconomic species, in the five large markets of Guinée Forestière in 2022, cited perceived reduced availability of certain plant species for medicines and food products (*Simbiano, 2022*). Considering that Guinea had already lost 92% of its total original forest before the end of the 20th century (*Sayer, Harcourt & Collins, 1992*) this use of habitat puts increasing pressure on the remaining natural resources. Additionally, in the Guinée Forestière region, a further 25% of the remaining area was lost between 2000 and 2018 (*Fitzgerald et al., 2021*). A large part of this forest loss can be explained by the strong growth in the average rate of agricultural expansion from 1.3% per year between 1975–2000 to 4.7% per year between 2000–2013, this is largely through slash and burn agriculture and often clearing original vegetation. However, this increased agricultural expansion was not distributed equally between regions (*CILSS, 2016*).

The threat register proposed here is based on the IUCN threat classification scheme Version 3.2 (*IUCN, 2012*) which provides a hierarchical structure of threat types for use in IUCN Red List assessments. This classification scheme was chosen as it is standardised and

internationally recognised, allowing comparisons with future datasets, and enables data to be easily incorporated into future Red List assessments.

## MATERIALS & METHODS

### Methodology

Initially, a paper questionnaire was formulated in Microsoft Word for data collection in the field, using 14 of the tier 2 IUCN threat categories (*IUCN, 2012*). A disturbance score of 'low', 'medium', 'high' or 'very high' was recorded for each threat with coordinates and a description of the threat. The timeframe was recorded according to if the threat was in the past, ongoing or poses a future threat.

However, this initial questionnaire did not gather precise enough data as the categories were too broad and the descriptions from the field team were not detailed enough, consequently, a different approach was developed.

A detailed Excel spreadsheet was prepared using the three-tier IUCN threat classification v.3.2 (*IUCN, 2012*). Since the authors developed the risk register, version 3.3 of the IUCN threat classification (*IUCN, 2022*) has been released and version 4 is likely to be released soon. This includes additional categories which we have not yet incorporated, but future editions of the register will be updated to be consistent with the new standard.

Since the release of version 3.3 in 2022, the scoring scheme IUCN used to quantify the impact of threats to a species has been temporarily suspended in the threats section of IUCN's assessment system (Species Information Service or SIS). In previous editions of SIS, timing of the threat (past, ongoing, future, unknown) was required, scope (how much of the population is affected) and severity (the impact of the threat) were used to calculate the threat; scope and severity were optional (*IUCN, 2022*). The IUCN scoring system considers scope *i.e.,* the amount of the population affected (minority, majority, whole). We omitted the scope aspect of the IUCN threat scoring since population data is rarely available for plants and therefore non-scientific estimates could introduce false bias into the scoring.

To simplify the data presentation in the spreadsheet (Supplementary Materials II), the tiers of the threat hierarchy have been grouped and can be collapsed to reduce the number of lines where specific threats are not triggered. There are three threat hierarchy classification columns, followed by columns for Location, Coordinates, Habitat and Description of activities. The next two columns have the scores for Disturbance (severity under IUCN) (1 = low to 4 = very high) and Timeframe (timing) (1 = past, 2 = future, 3 = ongoing) and a third column automatically calculates an overall Disturbance score by multiplying the disturbance and timeframe scores. A fourth, and last, column is dedicated to mitigation measures, either suggestions or actions already in place.

The overall disturbance scores are ranked 'low' to 'very high' in increments of three and colour coded according to RAG status (Red, Amber, Green), *i.e.,* a disturbance score of 1–3 is 'low', and therefore green, whereas a disturbance score between 10–12 would be 'very high', and therefore dark red.

We suggested that activities with a "low" score would require monitoring; activities with "medium" scores require monitoring and some mitigation; and activities with "high"

and "very high" scores require management interventions. For example, overcollection of non-timber forest products (NTFPs) such as collection of bark for medicinal purposes, recorded as a medium risk, local communities could be encouraged to put a harvesting quota in place, with supervision of a local committee. If forest clearance for poacher camps is recorded as a high risk, ecoguards would be required to patrol areas more frequently to apprehend or deter poachers.

The risk register format was transcribed into KoboToolbox (http://www.kobotoolbox.org, accessed on July 2021) to create a user-friendly format to record threats, using the ODK Collect smartphone application. ODK Collect automatically registers a geolocation for the threat and photos can be taken and associated with that datapoint.

Training sessions with ten Ecoguards from the five areas were held to introduce the form on ODK Collect and how to identify threats according to the IUCN threat categories. An initial "before- and-after session" was held to refine the data collection and discuss which categories best describe activities, to improve data quality. The Ecoguards subsequently went into the field in all five of the TIPAs to collect data on threats during their patrols, for five days. We did not prescribe a format of data sampling for this initial trial since the aim is that ecoguards will collect data during their routine patrols, with a picture building up over time. Data from all sites was collated through KoboToolbox into a spreadsheet and the datapoints mapped using QGIS 3.16 LTR. Quality control of the results was done by the first author, who translated the data into the risk register format in Excel. These registers were then shared with Centre Forestier N'Zérékoré and CEGENS who manage the Mt Nimba and Simandou areas. Risk registers for all five sites can be found in the supplementary materials and on the website of the National Herbarium of Guinea (http://www.herbierguinee.org).

## Study area

All five areas studies are documented Tropical Important Plant Areas (TIPAs) (*Couch et al., 2019*), Key Biodiversity Areas identified by *CEPF (2015)* and were also identified as protected areas by *Brugière & Kormos (2009)*. These areas were the focus of a CEPF funded project during which this pilot study took place. Many of these areas are designated as Classified Forests; however, in Guinea this does not equal protected area. Classified forests were originally designated for timber production and protection of water sources (*Brugière & Kormos, 2009*). It was later that some of these areas have been found to be important for conservation and incorporated into the Protected Area network.

Mount Béro is a Classified Forest of around 80 km$^2$ (*UNEP-WCMC and IUCN, 2021*) (central coordinates 8.200000°, −8.633333°), located mainly in the prefecture of N'Zérékoré, elevation starts at 600 m with the highest peak at 1,182 m. The area was classified in 1952, however, since 2009 there has been significant damage to the area; currently, the whole area is subject to development (*MEDD, 2022a*). Fourteen threatened plant species, including the world's largest population of two vulnerable species of massive flowering Acanthaceae (*Brachystephanus oreacanthus* and *Isoglossa dispersa*) and *Allophylus samoritourei* (EN) are found here. Submontane forest is present on the flanks of the mountain, and submontane grassland, of the type described as 'high altitude lateritic

bowé', at the summit, both are considered nationally threatened habitats (*Couch et al., 2019*).

Diécké's Classified Forest is the largest remaining area of lowland forest in Guinea (central coordinates: 7.525327°, −8.922195°), with an altitude span of 300 m to 550 m. It is located in Yomou prefecture, on the border of Liberia. It consists mainly of moist lowland forest with closed canopy. A total of 29 threatened plant species are found here, including many threatened trees (*Couch et al., 2019*). The forest has experienced logging in the past, but most of the core area of forest has remained intact with a closed canopy and open or shrubby undergrowth.

Ziama Massif Man and Biosphere Reserve, approx.111,000 ha, is in the prefecture of Macenta (central coordinates: 8.293572°, −9.344164°). It has an elevation span of 950 m to 1,400 m, with a highest point at 1,387 m. It was classified in 1942 and declared as a biosphere reserve in 1980. The Ziama Massif contains 33 threatened plant species and two endangered endemic plant species (*Gymnosiphon samoritoureanus* and *Inversodicraea pepehabai*) (*Couch et al., 2019*). It has significant dense submontane forest, lowland rainforest, swamp forest, gallery and secondary forest.

The southern Simandou mountains are situated in the south-east of Guinea (central coordinates: 8.538581°, −8.903452°) and includes the Classified Forest of Pic de Fon. They are part of the Loma-Man range that extends from western Ivory Coast into Sierra Leone. The highest peak, Pic de Fon, reaches 1,658 m. It has species associations with the Fouta Djallon Highlands and the Nimba Mountains. Over forty threatened plants have been recorded from the area. The ridges and flanks have a mosaic of submontane forest and high altitude lateritic bowé grassland with high species diversity, recognised as nationally threatened habitats of Guinea (*Couch et al., 2019*). A mining concession occupies part of the site, which threaten at least one species globally endemic to Pic de Fon.

The Nimba Mountains are situated in the south-east of Guinea, in the Lola Prefecture, extending into Liberia and Ivory Coast (central coordinates: 7.654659°, −8.387906°). Mt Richard-Molard, the highest peak in Guinea, reaches 1,752 m above sea-level. The Guinean part of the Nimba mountains covers 149.2 km$^2$ and has been protected since 1944. The majority (134.1 km$^2$) is recognised as a World Heritage Site (partly in Ivory Coast) and is a core area of the Nimba Mountains Biosphere Reserve, designated in 1980. The Guinean Nimba Mountains has 40 threatened plant species, of which at least six are globally endemic to Nimba. In 1993, an area of 15.16 km$^2$ was excised from the colonial Strict Nature Reserve for mineral exploration (*Brugière & Kormos, 2009*). There is currently an iron-ore mining concession of 6.25 km$^2$ in this area (*Couch et al., 2019*).

## RESULTS

Of the main threat cases identified, during the survey missions, according to tier 1 of the IUCN threat hierarchy 2. Agriculture and Aquaculture is by far the greatest threat (47.06%); this includes itinerant agriculture such as a field on a hillside which will be abandoned after harvest, small scale agriculture where cattle are used to plough floodplain areas for rice cultivation, or large-scale agriculture such as creation of a new plantation.

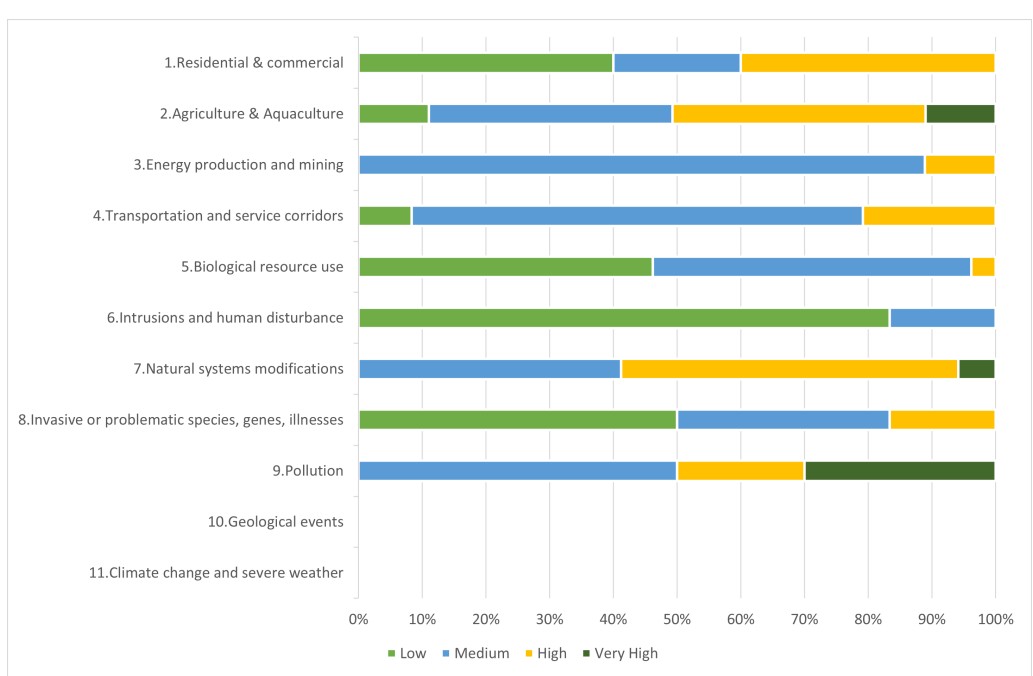

**Figure 2** Percentage of threat cases and their threat level per IUCN tier 1 threat class for all TIPAs.

The next biggest threat is 5. Biological resource use (17.99%) *e.g.*, overharvesting of tree bark, followed by 6. Human intrusions and disturbance (10.38%) (this could be poacher smoking racks or a camp), 4. Transportation and service corridors (8.30%) such as roads or pathways through the forest or mining roads, and 7. Natural systems modification (5.88%) such as an increase in frequency of human set fires (Fig. 2). The majority of threat cases (42.56%) were evaluated as medium, 26.30% were deemed a high threat with a quarter of cases recorded as low threat and only 6.57% evaluated as very high across all categories (Fig. 2).

A breakdown per site (Fig. 3) shows that Mont Béro and Southern Simandou Mountains (Pic de Fon) have the highest total number of threat cases, 102 and 73 respectively. Agriculture and Aquaculture class are the most important threat in all areas except in Ziama, where it is Biological resources use and Nimba where it is Human intrusions and disturbance (Table 1).

The distribution of threat cases recorded across the five sites can be seen in Fig. 4. The density of sampling varied across sites with Ziama, Nimba and Diécké being less well covered during the pilot survey than Mt Béro and Pic de Fon where there is better access for patrols.

Breaking this down further into the sub-categories, using Mont Béro as an example, the risk register (Material SI) shows that 65/68 threat cases recorded under 2. Agriculture & Aquaculture fall under sub-class 2.1) Agriculture & Perennial Non-Timber crops. The third sub-class shows that, at Mont Béro, these are a combination of 2.2.2) Small-scale agriculture (22), 2.2.3) Agro-industrial farming (37) and 2.2.1) Shifting agriculture (6)

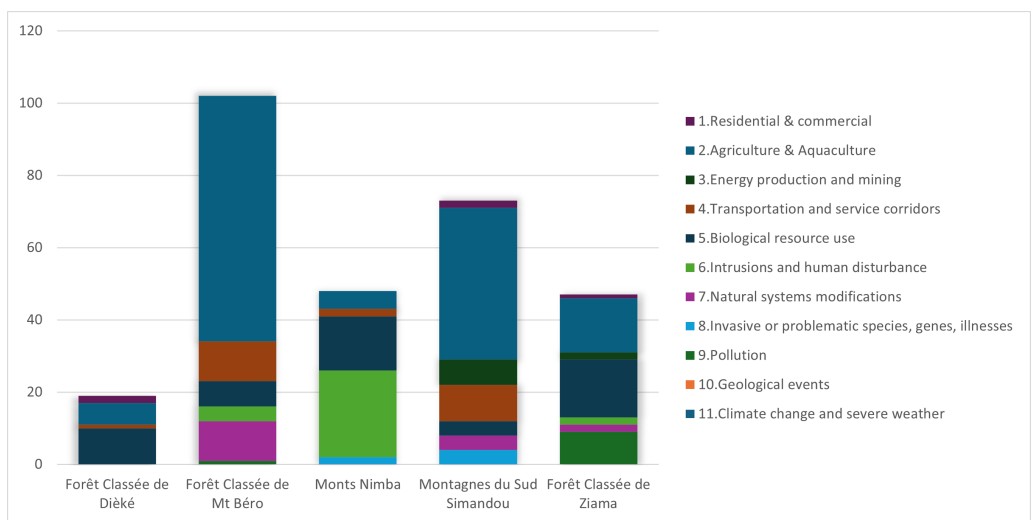

**Figure 3  Total number of threat cases recorded per TIPA per tier 1 IUCN threat category.**

and three records of grazing at various levels (Table 1). The majority of the agro-industrial farming at Mt Béro is plantations of coffee, oil palm or banana (see risk register in Supplemental Information). The RAG (Red, Amber, Green) status in the risk register shows that only five out of 65 agriculture threat cases were recorded as low risk, 41 as medium risk, 16 as high risk and three qualify as very high risk (Fig. 5). The low-risk areas are either abandoned or not yet fully established and are earmarked for removal by the ecoguards.

## DISCUSSION

This pilot study has resulted in the development of a useful tool to identify which threats are present in Tropical Important Plant Areas in Guinée-Forestière and how these threats are perceived by the ecoguards. Our data shows that agriculture is the main threat to forest loss in the Guinée-Forestière TIPAs, particularly Mt Béro (68/102) and Southern Simandou Mountains (45/77) (Fig. 3, Table 1, Material SI). This was evident in fieldwork undertaken in Diécké and Mt Béro in 2022, where the local communities have started to clear areas for cultivation within the boundary of the classified forests. This ground-truthed data confirms the remote sensing analysis by *Fitzgerald et al. (2021)*, who singled out Mont Béro as the area with the largest rate of deforestation in relation to area, primarily driven by subsistence agriculture. Mt Béro particularly suffered from lack of oversight by the authorities after the death of President Conté and the subsequent coup d'etat in 2009. Ecoguards patrolling this area were recalled and the protection of the area was left with the communities (CFZ to C Couch, 2021, pers.comm.). During the period 2009–2012, a large area of the forest was cleared by the local population, and in 2014, a road widening and upgrading scheme resulted in the loss of five trees of *Allophylus samoritourei* an EN species (*Cheek & Haba, 2016*). More recently, an area was being cleared for a banana plantation, but as a result

**Table 1  Total number of threats per IUCN threat class and per TIPA in Guinée-Forestiére.**

| IUCN threat class | Diécké | Mont Béro | Mont Nimba | Pic de Fon | Ziama | Total # of threat cases per class |
|---|---|---|---|---|---|---|
| **1. Residential & Commercial** | **2** | **0** |  | **2** | **1** | **5** |
| *1.1 Housing & Urban areas* | *2* |  |  | *2* | *1* |  |
| *1.2 Commercial & Industrial areas* |  |  |  |  |  |  |
| *1.3 Tourism & Recreational areas* |  |  |  |  |  |  |
| **2. Agriculture & Aquaculture** | **7** | **68** | **5** | **42** | **15** | **136** |
| *2.1 Agriculture & Perennial Non-Timber crops* | *7* | *65* | *5* | *41* | *15* | *133* |
| 2.1.1 Shifting agriculture | 4 | 6 | 1 | 23 | 1 | 35 |
| 2.1.2 Small-holder farming | 3 | 22 | 1 | 8 | 8 | 42 |
| 2.1.3 Agro-industry farming |  | 37 | 3 | 10 | 6 | 56 |
| *2.2 Wood & Pulp Plantations* |  |  |  |  |  |  |
| *2.3 Livestock Farming & Ranching* |  | *2* |  | *1* |  | *3* |
| 2.3.1 Nomadic grazing |  | 2 |  |  |  | 2 |
| 2.3.2 Small-holder Grazing, ranching or farming |  | 1 |  | 1 |  | 2 |
| 2.3.3 Agro-industry grazing, ranching or farming |  |  |  |  |  |  |
| *2.4 Marine & freshwater aquaculture* |  |  |  |  |  |  |
| **3. Energy Production & Mining** |  |  |  | **7** | **2** | **9** |
| *3.1 Oil & Gas drilling* |  |  |  |  |  |  |
| *3.2 Mining & Quarrying* |  |  |  | *7* | *2* | *9* |
| *3.3 Renewable Energy* |  |  |  |  |  |  |
| **4. Transportation & Service corridors** |  | **11** | **2** | **10** |  | **23** |
| *4.1 Roads & Railroads* |  | *11* | *2* | *10* |  | *23* |
| *4.2 Utility & Service Lines* |  |  |  |  |  |  |
| *4.3 Shipping lanes* |  |  |  |  |  |  |
| *4.4 Flight paths* |  |  |  |  |  |  |
| **5. Biological Resource Use** | **10** | **7** | **15** | **4** | **16** | **49** |
| *5.1 Hunting & Collecting Terrestrial animals* | *6* | *7* | *4* | *3* | *6* | *26* |
| 5.1.1 Intentional Use | 5 | 7 | 4 | 3 | 6 | 25 |
| 5.1.2 Unintentional Use | 1 |  |  |  |  | *1* |
| *5.2 Gathering Terrestrial Plants* |  |  | *9* | *1* | *5* | *15* |
| 5.2.1 Intentional Use |  |  | 9 | 1 | 5 | 15 |
| 5.2.2 Unintentional Use |  |  |  |  |  |  |
| *5.3 Logging & Wood harvesting* | *1* |  | *1* |  | *6* | *8* |
| 5.3.1 Intentional Use (Small scale) | 1 |  | 1 |  | 5 | *7* |
| 5.3.2 Intentional Use (large scale) |  |  |  |  |  |  |
| 5.3.3 Unintentional Use (small scale) |  |  |  |  |  |  |
| 5.3.4 Unintentional Use (large scale) |  |  |  |  |  |  |
| *5.4 Fishing & Harvesting Aquatic resources* | *3* |  |  |  |  | *3* |
| 5.4.1 Intentional Use (Small scale) | 3 |  |  |  |  | 3 |

| IUCN threat class | Diécké | Mont Béro | Mont Nimba | Pic de Fon | Ziama | Total # of threat cases per class |
|---|---|---|---|---|---|---|
| 5.4.2 Intentional Use (large scale) | | | | | | |
| 5.4.3 Unintentional Use (small scale) | | | | | | |
| 5.4.4 Unintentional Use (large scale) | | | | | | |
| **6. Human Intrusions & Disturbance** | | **4** | **24** | | **1** | ***37*** |
| *6.1 Recreational Activities* | | | | | | |
| *6.2 War & Civil unrest/ Military exercises* | | | | | | |
| *6.3 Work & other activities* | | *4* | *24* | | *1* | *37* |
| **7. Natural Systems Modifications** | | **10** | | **4** | **2** | ***16*** |
| *7.1 Fire & Fire Suppression* | | *10* | | *4* | *2* | *16* |
| 7.1.1 Increased Fire frequency/ intensity | | 10 | | 4 | 2 | 16 |
| 7.1.2 Suppression of fire frequency/ intensity | | | | | | |
| *7.2 Dams & Water Management/Use* | | | | | | |
| 7.2.1 Abstraction of surface water | | | | | | |
| 7.2.2 Abstraction of ground water | | | | | | |
| 7.2.3 Small Dams | | | | | | |
| 7.2.4 Large Dams | | | | | | |
| *7.3 Other Ecosystem Modifications* | | | | | | |
| **8. Invasive or problematic species, genes, diseases** | | | **2** | **4** | | **6** |
| *8.1 Invasive Non-Native/Alien Species* | | | *2* | *4* | | *6* |
| *8.2 Problematic Native Species* | | | | | | |
| **9. Pollution** | | **1** | | | **9** | ***10*** |
| *9.1 Domestic & Urban wase water* | | | | | | |
| *9.2 Industrial & Military effluents* | | | | | | |
| 9.2.1 Oil spills | | | | | | |
| 9.2.2 Seepage from Mining | | | | | | |
| *9.3 Agricultural & Forestry effluents* | | *1* | | | *3* | *4* |
| *9.4 Garbage & solid waste* | | | | | *6* | *6* |
| *9.5 Airborne pollutants* | | | | | | |
| 9.5.1 Acid rain, smog, ozone | | | | | | |
| **10. Geological Events** | | | | | | |
| *10.1 Volcanoes* | | | | | | |
| *10.2 Earthquakes* | | | | | | |
| *10.3 Avalanches & Landslides* | | | | | | |
| **11. Climate change & severe weather** | | | | | | |
| *11.1 Habitat shifting alteration* | | | | | | |
| *11.2 Drought* | | | | | | |
| *11.3 Temperature extremes* | | | | | | |
| *11.4 Storms & Flooding* | | | | | | |
| *11.5 Other* | | | | | | |

**Notes.**
Numbers in bold denote total number of threats per tier 1 threat category.

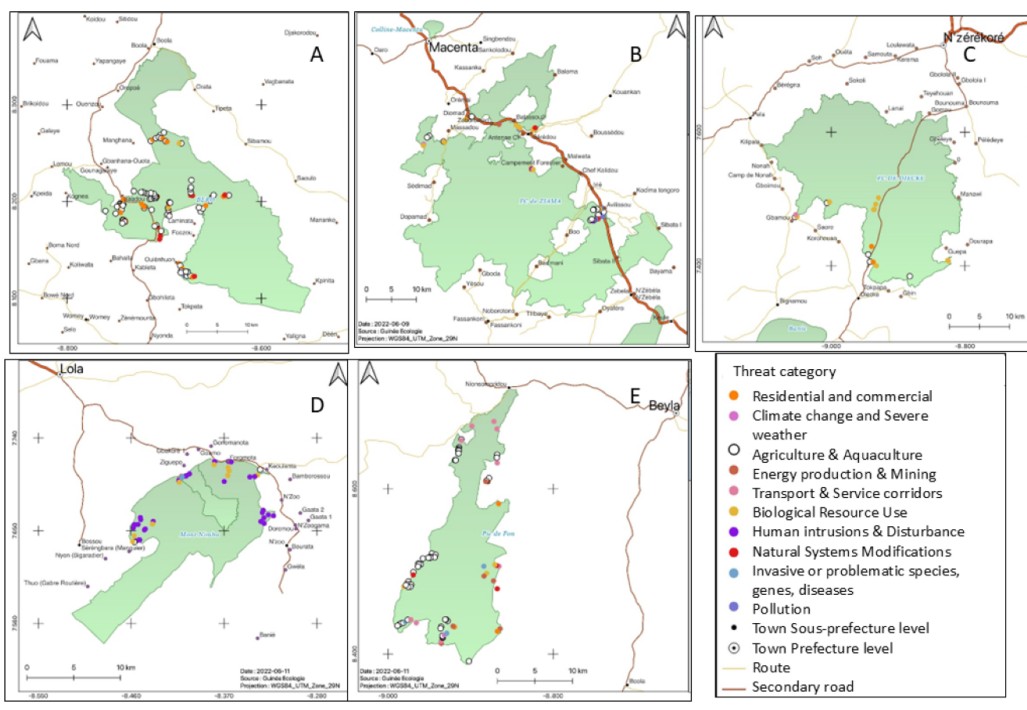

**Figure 4 Maps depicting the different threat types and their location at the five TIPAs.** (A) Mt Béro, (B) Ziama, (C) Diecke, (D) Monts Nimba, (E) Southern Simandou Mountains (Pic de Fon).

of the risk register fieldwork, this was discovered, and the people brought before the local authorities.

The history of protected areas in Guinea have been documented to some extent by *Brugière & Kormos (2009)* and *IUCN/PACO (2008)* however, successive governments and military coups have caused disruption in both the level of protection and designation of new areas. Changes in name of the organisations and which ministries they come under and how they work together has also caused issues with jurisdiction. In Guinée Forestière, as a result, there are two different offices with conservators/ecoguards. CEGENS manages Mt Nimba and Pic de Fon, CFZ manages Diécké, Ziama, Mt Béro, but OGPNRF (the national parks and fauna reserves department) manages other areas. However, then there are the prefectorial level environment offices which have other jurisdictions, and these can come into conflict with the conservators/ecoguards. *Brugière & Kormos (2009)* identified 16 KBAs in Guinea in 2009 through an exercise of applying the KBA criteria using mammals as a proxy for biodiversity. Only half of the areas identified correlated to Important Bird Areas, identified by Birdlife International and similarly TIPAs do not necessarily overlap with either designation. However, at the time of their study knowledge of Guinean plant diversity was more incomplete than at present (*Gosline et al., 2023*). The new proposed protected areas network will take into account many of the TIPAs since the government committed to protecting TIPAs when they were presented in 2019 (*Couch et al., 2023*).

Important Plant Area (IPA) designation follows three criteria: (1) threatened species, (2) botanical richness and (3) threatened habitats (*Darbyshire et al., 2017*). Many tropical

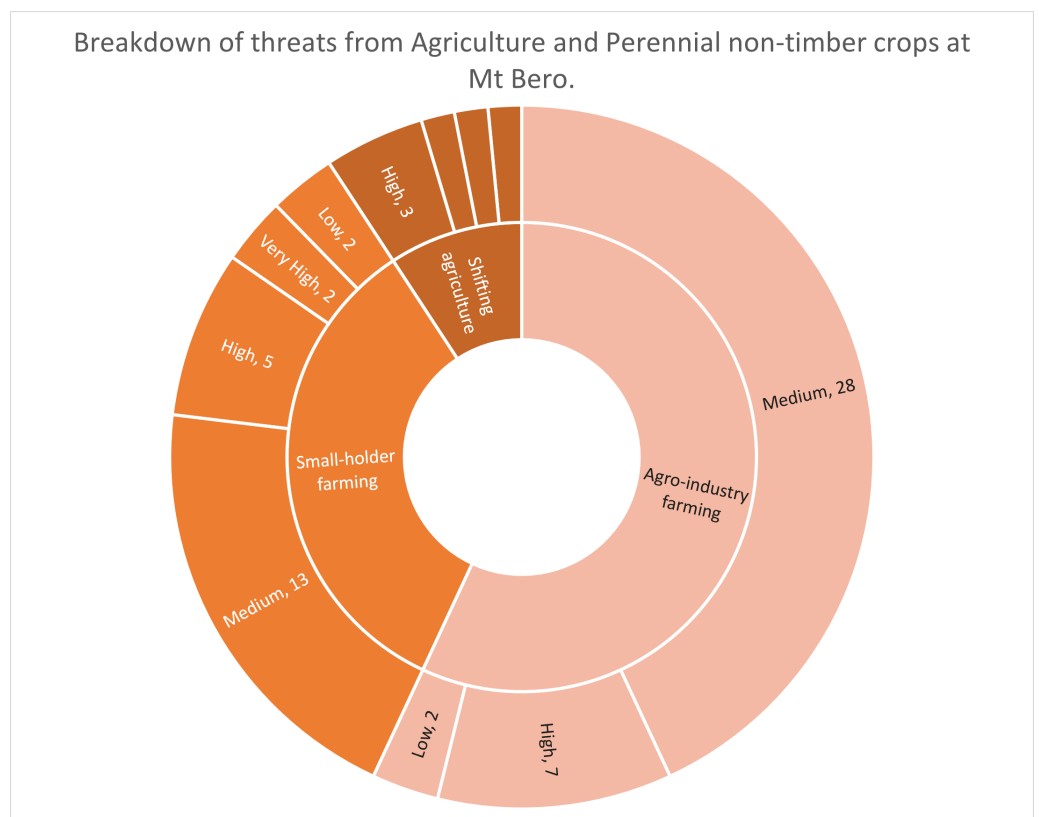

**Figure 5** Diagram showing the proportion of threats and their RAG status at Mt Béro for threat class 2.1 Agriculture and Aquaculture subclasses.

countries lack the data to assess their threatened plant species either data is disparate, in unpublished reports and grey literature or there is no database of records even if there are herbaria. Guinea still lacks sufficient data to be able to assess species at a national level, so the majority of IUCN Red List assessments were done at a global scale. This lack of data coverage is not uncommon and many of the counties in the Kew TIPA programme have had to undertake red listing activities initially to know what and where their threatened species are before being able to identify areas to protect. Mozambique (*Darbyshire et al., 2019*; *Darbyshire et al., 2023*; *Odorico et al., 2022*), Bolivia (*Moraes, Maldonado & Zenteno-Ruiz, 2018*) and British Virgin Islands (*Barrios et al., 2019*) had to focus on plant assessment before using criterion A to identify TIPAs.

As part of the process to designate TIPAs in Guinea (2016–2019), over 200 plant species were assessed for the IUCN Red List. When conducting initial screening in 2016 for plant species already assessed on the IUCN Red List only 66 assessments of threatened plants were recorded (M Cheek, 2024, pers. obs.). It was noticeable that many West African timber species, assessed under the old criteria, *e.g.*, *Entandrophragma cylindricum* (*Hawthorne, 1998a*), *Khaya anthotheca* (*Hawthorne, 1998b*) were not listed as occurring in Guinea, even though they are present. This is likely because data on Guinean plants has not been available
in a digital format on a global database *e.g.*, GBIF, the Flore des Angiospermes de Guinée (Lisowski) was only published in 2009 and many herbaria had only just started to digitise their specimens under project such as the African Plants Initiative (later incorporated into JSTOR plants). Red listing done by RBG Kew, Missouri Botanical Gardens and the Global Trees Initiative (BGCI) has addressed this issue, though the number of threatened species on the IUCN Red List still doesn't match the total known threatened plants of Guinea (230 *vs* 300) (*IUCN, 2022*), due to backlogs in publishing. These older assessments also lacked any meaningful threat data. Only because of recent field work since 2005 has detail about threats been collected, often during Environmental Impact Assessment studies. Fieldwork to identify Tropical Important Plant Areas (2016–2019) covered a large part of the country and enabled researchers to understand what the threats are and the extent of these on the flora, enabling more comprehensive Red List assessments. During fieldwork for EIAs and TIPAs, c. 30 new species to science have been described and assessed as threatened (*Couch et al., 2023*; *Gosline et al., 2023*).

The Southern Simandou Mountains (Pic de Fon) showed additional threats relating to the pre-mining activities in the area, particularly road building and introduction of invasive plants. Pre-mining activities have occurred since 2005 but has gone through periods of high and low activity. Similarly, these threats have been recorded at Nimba, but to a lesser extent since the mining concession is smaller.

All areas show that forest resources are harvested as NTFPs, with some being more impacted than others. Sustainable harvesting methods need to be explored with local communities (Material SI). There is a general attitude that the forest will always be there for people to exploit. Awareness training with local communities has already brought a better understanding of the role that the forest plays in not only providing useful products, but also its role in managing local climatic conditions (FJ Simbiano, 2023, pers. comm.). Currently, we are working with several communities to install plant nurseries for threatened and useful plant species to promote conservation and rehabilitation of these forests (*Simbiano, Mamy & Onivogui, 2023*).

The use of a four-point RAG scale for determining the level of threat was helpful to maintain consistency, though opinions of perceived threats can differ from one area to another. Further training on threats and how they are presented and classified according to the IUCN threat hierarchy will be needed to ensure consistency across TIPAs. Through the training exercises it was noted that some exploitation of particular species was recorded as a threat when in fact it is not, since the species concerned, *Harungana madagascariensis* Lam. ex Poir. (Hypericaceae) is a pan-African pioneer and grows in a variety of habitats. Therefore, this could be termed sustainable use, since only a few stems were extracted. Equally, the threat of unsustainable harvesting of *Raphia hookeri* G.Mann & H.Wendl. (Arecaceae), ''raffia palm'', and clearance around these trees, needs better defining to understand the threat processes. Thus, registering and repeated monitoring at sites could be used to gain a deeper understanding of the use of species and habitats by local communities.

The tool outlined in this paper can be used for all threats to habitats and species, not just those pertaining to the forest/plant species elements as was the focus here. Data on plant species as mentioned earlier is often disparate, but quantitative data specifically

on threats to plants outside of the wider commercial use can be hard to find if it exists at all. It is intended that this could provide a simple method for ecoguards to monitor and manage threats to plant species within TIPAs and other protected areas. All those involved in the pilot study felt that it was a useful tool and could be used for monitoring as well as registering threats, if a suitable database was created to store and access the data. There is currently no system across the PA network in Guinea that is being used to record quantitative data on threats to plants or more generally. The creation of a database is part of follow-on funding secured until 2026. The database is being developed in collaboration with Université Gamal Abdel Nasser de Conakry. The database will have the functionality to produce a user-friendly report of the risk register. Monitoring of threats will be done by resurveying the same areas over time to see if there is a reduction in the RAG status *i.e.,* more activities registered as green, than amber or red following implementation of effective mitigation measures. This will be visible when a new register is generated. Moreover, this quantified threat data can directly feed into IUCN Red List assessments at national, regional or global levels, providing more accuracy and detail on conservation measures and research required. Currently, Guinea does not have sufficient geographic distribution data to conduct national plant Red List assessments, however, the risk register data will contribute to future assessments. The risk register data can also be applied to assessments for the Red List of Ecosystems which requires a review of threats to an ecosystem during the evaluation process (*IUCN, 2016b*). Thus, our data can assist with future red listing efforts of species and ecosystems both nationally and globally. Using the same system for recording threats will facilitate comparison between TIPAs, countries and projects. We think this could be relevant to other projects across West Africa or globally which are seeking to monitor threats to plants in their research areas.

The authors are unaware of other studies using a risk register approach to record and monitor threats to plants and the wider landscape (*e.g.*, TIPAs or KBAs) in other African countries. A recent update to the Management Effectiveness Tracking Tool (METT) (*Stolton & Dudley, 2016*; *Stolton, Hockings & Dudley, 2020*) now includes a datasheet using the IUCN threat hierarchy to assess threats which our data can directly feed into, if METT analysis is performed on any of the study areas. The Integrated Management Effectiveness Tool or IMET system uses a threat calculator, but it is unclear if a standardised list of threats is used (*Paolini & Rakotobe, 2023*).

Protected Area Management Effectiveness (PAME) systems such as RAPPAM (*Ervin, 2003*; *IUCN/PACO, 2008*) or Priority Threat Management (*Carwardine et al., 2019*) are often done through interviews with protected area managers, stakeholders, with spatial analysis *etc.* This requires a level of existing knowledge about the threats which may not exist quantitatively. The analysis of protected areas in Guinea in 2008 by IUCN used the RAPPAM methodology and mentions that a limiting factor is the knowledge of the participants. IUCN/PACO hoped by having a sufficient cross section of stakeholders that this limitation would be offset.

Many see the METT and other PAME tools as separate to the day-to-day management and the outcomes are not always integrated afterwards (*Bialowolski et al., 2023*). The risk

register developed here is designed to be used on the ground for the management of threats-identifying and then implementing mitigation measures.

Threats to mammals or birds may be more obvious and therefore better recorded than those for plants which all too often get lumped as 'deforestation' or 'habitat degradation' but are not well defined and could be affecting some already threatened species more than others. *Battisti, Poeta & Fanelli (2016)* looked at different rating typologies. They made a distinction between a relative approach *i.e.,* all threats are simultaneously evaluated with respect to one another, not independently; and an absolute approach or threat by threat approach which looks at the impact of individual threats on targets. The relative approach may be useful for higher level *e.g.*, regional or national park management. For areas of particular conservation interest, such as TIPAs, our absolute approach detailing classified threats that have been mapped, quantified and monitored can provide insights into where management interventions are most needed for areas of high plant diversity.

## CONCLUSIONS

This study has confirmed that there are significant threats to plants in TIPAs of Guinée Forestière, supporting the results of *Fitzgerald et al. (2021)* who identified agriculture as the most significant threat. The threat risk register is easy to use, by gathering data using ODK Collect and the Excel format can provide a simple way to present the data, though this would be more efficient if it can be automatically generated from the database currently in development. Our approach can be used more widely across TIPAs or Protected Areas networks to record and monitor threats to plants and the wider landscape using a system that is comparable across areas and countries. The data required will be useful for national and regional level Red List species and ecosystem assessments and particularly for those in Guinea in the future. It will also raise awareness of plant specific threats among ecoguards/conservators by identifying significant threats to threatened or useful plant species, not just wood cutting and harvesting of NTFPs, and identify where interventions are needed. This information can feed into a variety of other assessment processes once available through an accessible database.

## ACKNOWLEDGEMENTS

Our thanks to Kaman Guilavogui (Herbier National de Guinée) for his assistance with training, the conservation staff of Centre Forestier N'Zérékoré and Office Guinean des Parcs Nationaux et Reserves de Faunes (OGPRNF), and the local communities around the sites at Mt Béro, Diécké, Nimba, Ziama and Pic de Fon for their collaboration. Ana Rita Simões and Xander van der Burgt for editing the manuscript.

### Funding

This work was supported by a Critical Ecosystem Partnership Fund (CEPF) grant (No. CEPF-110531). The funders had no role in study design, data collection and analysis, decision to publish, or preparation of the manuscript.

### Grant Disclosures

The following grant information was disclosed by the authors:
A Critical Ecosystem Partnership Fund (CEPF): CEPF-110531.

### Competing Interests

The authors declare there are no competing interests.

### Author Contributions

- Charlotte Couch conceived and designed the experiments, performed the experiments, analyzed the data, prepared figures and/or tables, authored or reviewed drafts of the article, and approved the final draft.
- Faya Julien Simbiano performed the experiments, authored or reviewed drafts of the article, and approved the final draft.
- Mamadou Diawara conceived and designed the experiments, analyzed the data, authored or reviewed drafts of the article, and approved the final draft.
- Edgar François Loua conceived and designed the experiments, analyzed the data, prepared figures and/or tables, authored or reviewed drafts of the article, and approved the final draft.
- Léonce Mamy performed the experiments, authored or reviewed drafts of the article, and approved the final draft.
- Martin Cheek analyzed the data, authored or reviewed drafts of the article, and approved the final draft.
- Sékou Magassouba conceived and designed the experiments, authored or reviewed drafts of the article, and approved the final draft.

### Data Availability

Raw data is available in the Supplemental Files.

### Supplemental Information

Supplemental information for this article can be found online at http://dx.doi.org/10.7717/peerj.19629#supplemental-information.

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
