# Peer review of "Developing a threat risk register based on the IUCN threat hierarchy for five tropical Important Plant Areas in Guinea"

_PeerJ, doi:10.7717/peerj.19629_

## Round 0.1 · original submission · Major Revisions

· Academic Editor

Major Revisions

Both reviewers were globally positive on your paper. They made however important comments and suggestions. Please consider all of them when elaborating a revised version and provide point-by-point responses.

Reviewer 1 ·

Basic reporting

The main text is written in good English and is well-structured. The threat risk register presented in this paper is put well into context. Figures and tables are fine in form and helpful to understand. Raw data are supplied.

Experimental design

This is not an experimental study but rather a presentation of a tool for conservation assessments. Therefore the criteria for this section can not be applied. The results presented here are evaluations of the data collected by ecoguards in the 5 TIPAs. As a demonstration of the tool they are totally fine, but they don't seem to follow any systematic sampling scheme or at least it is not mentioned in the text.

Validity of the findings

The results demonstrate the easy use of the tools provided.

Additional comments

Study Area
Maybe the classified forest category should be briefly explained, as people not working in the region may not be familiar with it.

Methodology
154ff: The scheme used here is based upon the IUCN threats classification scheme 3.2, that seems to be superceded by version 3.3. So I would expect some remarks how the threat register relates to the new classification scheme. I did not have any access to version 3.2, but I see that there are some differences in calculating a total impact (IUCN: scope x severity x timing vs this paper: risk score x timing).
163ff: RAG is red-amber-green? a look in the supplementary material helps, but a short explanation would be better here.
185: Centre Forestier (without accent and e)

Results
Maybe better put the IUCN categories with numbers in quotes, in this form it is quite confusing to read.
I mainly understood the developed tool as the result, as indicated by the title. For a meaningful evaluation of the data I would expect (1) a sampling scheme that ensures a good geographical and ecological coverage of all 5 areas, and (2) an explanation how comparability of assessments by different ecoguards has been achieved.
Figure 2 is not quite clear to me. Are threats of different impact levels all summarized in a single bar? Then stacked bars of different levels would give a better picture, the same applies to figure 3 that could also provide more details showing different threat classes.

Discussion
251ff: Some more details on the availability of distribution data and a more general discussion of red-listing in data-poor countries may be interesting. AOO/EOO-based assessments in other countries also showed potential, but actually lacked the thorough analysis of threats.

·

Basic reporting

English is not the first language of the reviewer, no comments on that part.

For effective protection of officially protected vegetation you need to collect a lot of data, part of that data is about the threats. Threats are changing through time and need careful monitoring. To make this as easy as possible you need practical tools. It is useful to publish about the progress in this field like the pilot project that is described in detail in this manuscript.

This manuscript describes a way of measuring the threats and the damage to protected forest vegetation and how to share this information with the world. Of course this is a necessary base for better protection. The authors are convinced that their way of working is better than how this was done before. To convince the reader they should maybe write in a bit more detail about the history of these threats and the threat management. They should not expect that everything in the References is already known to the reader.
I would suggest changing the structure of the paper and put the pilot study/Risk Register first (also see my comments in the PDF). The threat measurement tools and not the 5 TIPA’s are the subject of this paper; the 5 TIPA’s are used to explain the usefulness of the tools. Of course it is still important that these TIPA’s are described in detail like they have done already. It is clear that the 5 TIPA’s in Guinea are chosen as example because that is where the research of the authors was already focussed on. This can be concluded from References but should be easier to find back in the paper as well.
Some of these protected areas are already officially protected for tens of years. What were the earlier threats and ways of protection? There must be a bit more to tell about the earlier threats and management especially when you want to explain the need for new and better tools.

See my other comments in the PDF

Experimental design

no comment

Validity of the findings

Future use of these tools by monitoring of protected vegetation will have to show their usefulness.

Additional comments

See my comments in the PDF.

Reviewer 3 ·

Basic reporting

See attachment.

Experimental design

See attachment.

Validity of the findings

See attachment.

Annotated reviews are not available for download in order to protect the identity of reviewers who chose to remain anonymous.

---

## Round 0.2 · Minor Revisions

· Academic Editor

Minor Revisions

Dear authors, please consider the latest suggestions provided by the reviewer and modify the manuscript accordingly, or specify the reason why they were not accepted.

Please note that the reviewer has highlighted some requests in the attached PDF file.

I'm looking forward to receiving the revised manuscript.

·

Basic reporting

English is not the first language of the reviewer, no comments on that part.

Experimental design

As for first version.

Validity of the findings

As for first version.

Additional comments

I have checked the new version of “Developing a threat Risk Register based on the IUCN
threat hierarchy for five Tropical Important Plant Areas in Guinea” (#90504). I am happy to see that most of my earlier comments could be used for this new version.
Some part of the new text have clearly been written under some time pressure, I have added a few comments where the text can use a bit more attention of the authors. Otherwise I am happy with the new version.
See my comments in the PDF.

---

## Round 0.3 · accepted · Accept

· Academic Editor

Accept

Dear Authors,

The manuscript has been revised according to the latest minor indications and therefore can be accepted.

Best regards